# Improving End-To-End Autonomous Driving with Synthetic Data from Latent Diffusion Models

Harsh Goel
University of Texas Austin
harshg99@utexas.edu

Sai Shankar Narasimhan
University of Texas Austin
nsaishankar@utexas.edu

## Abstract

*The autonomous driving field has seen notable progress in segmentation and planning model performance, driven by extensive datasets and innovative architectures. Yet, these models often struggle when encountering rare subgroups, such as rainy conditions. Obtaining the necessary large and diverse datasets to improve generalization in these subgroups is further hindered by the manual annotation's high cost and effort. To tackle this, we introduce SynDiff-AD, a unique data generation pipeline designed to synthesize realistic images for under-represented subgroups. Our system utilizes latent diffusion models (LDMs) with meticulous text prompts to generate images from existing dataset annotations, faithfully preserving their semantic structure. This crucially eliminates the need for manual labeling. By augmenting the original dataset with images generated by our system, we demonstrate the improved performance of advanced segmentation models like Mask2Former and SegFormer by +1.4 mean Intersection over Union (mIoU). We also observe enhanced driving capabilities in end-to-end autonomous planning models like AIM-2D and AIM-BEV across diverse conditions by over 20%. Our thorough analysis highlights that our method also contributes to overall model improvement.*

## 1. Introduction

The field of Autonomous Driving (AD) has made significant strides in recent years, fueled by the massive datasets collected by vehicle fleets. This data explosion offers invaluable training resources. However, a key issue emerges: the collected data often exhibits a bias towards favorable weather conditions, such as sunny and clear skies. Datasets like the Waymo Open Dataset [34] and BDD100K [41] illustrate this, with a substantial portion of their samples falling into this category. This imbalance impacts model performance on segmentation [17] and planning tasks [1, 2] when encountering less common scenarios like rain or clouds. Furthermore, the associated manual labeling costs

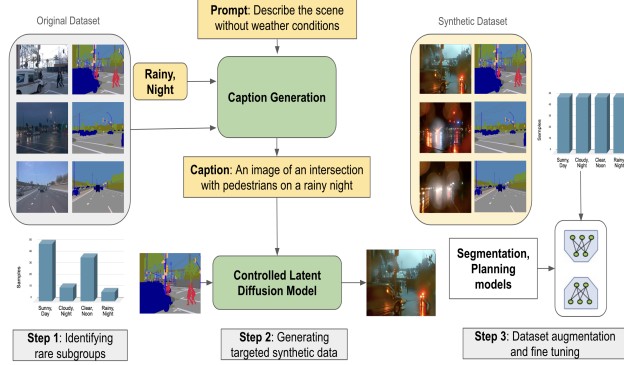

Figure 1. **Our proposed synthetic data generation method SynDiff-AD.** We begin by identifying rare subgroups in the dataset. We propose a sophisticated caption generation scheme to direct image generation with ControlNet towards rare subgroups.

are substantial for collected extra data.

To address these challenges, we propose SynDiff-AD, a novel pipeline to generate realistic data for under-represented subgroups, to augment the existing AD datasets. SynDiff-AD leverages the vast improvements in the field of text-to-image controlled generation such as [28, 30] to generate variants of the existing dataset samples while preserving their semantic structure. Our approach outlined in Fig. 1, eliminates the need for manual labeling in semantic segmentation and end-to-end autonomous driving tasks. First, we pinpoint under-represented subgroups within a dataset. We then employ ControlNet [42], a latent diffusion model (LDM), to transform samples from over-represented subgroups into the desired under-represented conditions while safeguarding their semantic content. The generated images are used to augment the existing AD dataset, allowing fine-tuning of segmentation and end-to-end (E2E) driving models. Crucially, our approach demands no human labeling effort.

Our key contributions are summarized as follows:

1. We present SynDiff-AD, a conditional LDM-based method inspired by ControlNet that synthesizes samples for under-represented dataset subgroups. Our sys-

tem modifies images from over-represented subgroups, meticulously preserving semantic structure.

2. Drawing inspiration from textual inversion concepts, we design a novel prompting scheme for data augmentation. We leverage LLaVA, a vision-language captioning model, to describe images within over-represented subgroups. These descriptions are carefully modified to reflect under-represented scenarios, guiding the generation process in our system.

3. Our extensive experiments show that fine-tuning state-of-the-art semantic segmentation and autonomous driving models with the augmented dataset yields improved average performance across all subgroups.

## 2. Related Work

### 2.1. Semantic Segmentation and End-to-End Autonomous Driving

Prior research in semantic segmentation and E2E AD has primarily emphasized architectural modifications. State-of-the-art semantic segmentation models [5, 6, 23, 38] now heavily utilize the Transformer architecture [36]. In E2E AD, the focus has been on designing model architectures to predict future waypoints for vehicle tracking, often drawing on multimodal data like camera images and LiDAR depth information [4, 8, 9, 25, 32]. In contrast, our paper explores a different direction: improving the performance of state-of-the-art semantic segmentation and E2E AD models through synthetic data augmentation.

### 2.2. Generative Models to Improve Synthetic Data for Perception and Planning tasks

Text-to-image generative models built on Diffusion Probabilistic Models (DPMs), such as Stable Diffusion [30] and Glide [26], excel at generating high-quality synthetic data from text. However, they struggle with precise modifications of objects within an image. Solutions have emerged, including Pivotal Tuning [29] for targeted adjustments, Textual Inversion [12, 31] for adapting images to new concepts, and ControlNet [42] for conditioning generation on elements like semantic maps. FreestyleNet [39] further refines control through attention mapping, while Edit-Anything [13] combines tools like ControlNet, Segment Anything (SAM) [18], and BLIP2 [20] for versatile editing. These methods have been adapted to improve perception models that is complementary to GAN-generated synthetic data to improve perception models [11, 43]. Diffusion models have expanded synthetic data applications to object classification [3, 15], detection [21], and segmentation [16, 19, 35, 37, 40, 44]. Methods like DatasetDM [37] generate both images and annotations, and DGInStyle [16] focuses on style transfer. However, none specifically address segmentation performance on under-represented sub-

groups. For autonomous planning, synthetic data traditionally comes from expensive 3D game engines (Unity, UE4), facilitating procedural generation and expert data collection [33]. In contrast, our approach leverages existing semantic data for image generation, eliminating the need for additional human expert input.

## 3. Method

We outline our proposed approach SynDiff-AD, in this section. The problem statement is described in Section 3.1. Our method for generating synthetic data conditioned on semantic masks for under-represented subgroups is detailed in Section 3.2. We enhance the quality of generated data fine-tuning ControlNet using text captions and style prompting (Section 3.3). Finally, Section 3.4 outlines the procedure to synthesize data targeted to under-represented subgroups for semantic segmentation and autonomous driving tasks.

### 3.1. Preliminaries

Our goal is to enhance the performance $\mathcal{P}_f(z; \theta) = \mathbb{E}_{p_{\mathcal{D}}(\mathbf{x}, \mathbf{y}|z)} L(f(\mathbf{x}; \theta), \mathbf{y})$ of a model $f$ parameterized by $\theta$ across each subgroup $z$ distributed in the dataset $\mathcal{D}$ by $p_{\mathcal{D}}(z)$. Additionally $p_{\mathcal{D}}(\mathbf{x}, \mathbf{y}|z)$ represents the conditional data distribution corresponding to subgroup $z$. In this paper, $f(\mathbf{x}; \theta)$ refers to a parameterized segmentation or autonomous driving model that produces an output $\mathbf{y}$ for input image(s) $\mathbf{x}$. For segmentation, $\mathbf{y}$ is a segmentation map and for E2E AD, $\mathbf{y}$ denotes the position offset. Additionally, $L(f(\mathbf{x}; \theta), \mathbf{y})$ is a metric which is the Mean Intersection over Union (mIoU) and normalized driving score [8, 9] for segmentation and E2E AD respectively.

We assume a predefined set of subgroups $\mathcal{Z}$ which comprise an *operational design domain* [10]. Each subgroup within $\mathcal{Z}$ is a semantic attribute, for instance weather or lighting conditions. The domain is compositional [24], i.e. $\mathcal{Z} = \mathcal{Z}_0 \times \mathcal{Z}_1 \times .... \times \mathcal{Z}_{n_Z-1}$, where each $\mathcal{Z}_i$ represents a distinct semantic dimension. Here, $n_Z$ denotes the total number of semantic dimensions. We use CLIP [27] to partition existing test datasets based on these semantic attributes, as they lack explicit subgroup labels. The evaluation uses corresponding training images if a subgroup is under-represented in the test set.

### 3.2. Label Conditioned Image Generation

SynDiff-AD leverages semantic masks with text-to-image generative models to synthesize pixel-aligned image-mask pairs for under-represented subgroups. We utilize ControlNet [42], which extends Stable Diffusion [30] by providing additional control, like semantic masks, during the Latent Diffusion Model's (LDM) denoising process. This is achieved through a mirrored U-Net encoder that guides image generation while maintaining the semantic mask's layout. When fine-tuning ControlNet for segmentation or au-

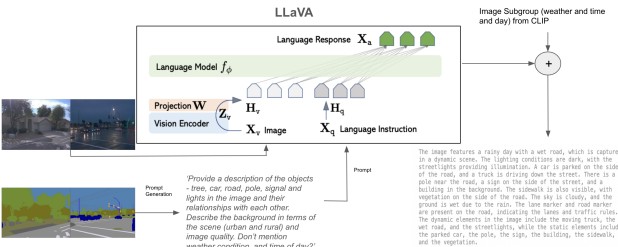

**LLaVA**

Figure 2. **Caption Generation Pipeline.** Our proposed caption generation scheme where we query a VLM such as LLAVA (image reproduced from the paper) with an image in the dataset, and a prompt constructed from the semantic masks to obtain a caption.

tonomous driving, we freeze the LDM parameters. Training focuses on ControlNet's mirrored U-Net, which receives semantic masks. Text prompts are used to describe the image, incorporating the desired subgroup as a style. (See Sections 3.3 for details). We fine-tune ControlNet separately for each task (segmentation, driving).

### 3.3. Improving text prompts via VLMs and Subgroups as style prompts for ControlNet

To improve synthetic data quality, we use high-quality captions as synthesis prompts. These captions, previously shown to enhance image realism [3, 13], are generated with LLaVA [22] in our work. LLaVA leverages a VLM (Vicuna [7]) with CLIP-extracted image features to produce captions. In SynDiff-AD, we generate captions for segmentation and driving datasets using LLaVA with a visual-questioning format. Queries are based on semantic mask classes, and we instruct LLaVA to exclude keywords related to target subgroups $z \in \mathcal{Z}$. This process is illustrated in Fig 2. Experiments in the Appendix demonstrate that LLaVA captions improve the realism of ControlNet synthesized images for segmentation tasks. Additionally, we use style prompting to ensure ControlNet captures the unique characteristics of each subgroup. Specifically, we augment the text prompts (generated as described in Section 3.3) with the subgroup $z$ (identified by CLIP) and its semantic dimension. For example, if the dataset is partitioned by weather, we append "Image taken in Cloudy weather" to LLaVA's generated caption. This style prompting directs ControlNet to synthesize data tailored to specific under-represented subgroups.

### 3.4. Generating Synthetic Dataset

**Synthesizing images with ControlNet for Segmentation and Autonomous Driving**

We generate synthetic images for semantic segmentation using Algorithm 1. For autonomous driving, we make a key modification. Since driving annotations (e.g., waypoints) don't describe the visual scene, we need a semantic representation to ensure realistic synthesis. We obtain this lay-

---

**Algorithm 1** Synthetic Data Augmentation Algorithm

---

1: **Input:** Dataset $\mathcal{D} = \{(\mathbf{x}, \mathbf{y}, z)\}^N$ with images $\mathbf{x}$, masks $\mathbf{y}$, and annotated subgroups $z$, ControlNet $g_\theta(\mathbf{x}|\mathbf{y}, c)$ where $c$ is a text caption, VLM Model - $VLM(\mathbf{x}, p)$ where $p$ is a prompt, $n_{\text{synth}}$ number of synthetic images

2: **Output:** Augmented Dataset $\mathcal{D}_{\text{aug}}$

3: **procedure** DATAAUGMENTATION($\mathcal{D}$)

4:     $\mathcal{D}_{\text{synth}} \leftarrow \{\}$

5:     **for** $i \leftarrow 1$ to $n_{\text{synth}}$ **do**

6:         Sample $z \sim p_\mathcal{D}(z)$ `// Sample subgroup from dataset`

7:         Sample $(\mathbf{x}, \mathbf{y}) \sim p_\mathcal{D}(\mathbf{x}, \mathbf{y}|z)$ `// Sample image and mask given source subgroup from dataset`

8:         Obtain prompt $p$ based on semantic label $y$ and subgroup $z$

9:         $c \leftarrow VLM(\mathbf{x}, p)$ `// Caption c`

10:        Target subgroup $z_t \sim U(\mathcal{Z})$ `// Sample target subgroup uniformly`

11:        $c_t \leftarrow c \cup z_t$ `// Add target subgroup as style`

12:        Generate synthetic image $\mathbf{x}_{\text{synth}} \sim g_\theta(\mathbf{x}|\mathbf{y}, c_t)$

13:        $\mathcal{D}_{\text{synth}} \leftarrow \mathcal{D}_{\text{synth}} \cup \{(\mathbf{x}_{\text{synth}}, \mathbf{y})\}$

14:     **end for**

15:     $\mathcal{D}_{\text{aug}} \leftarrow \mathcal{D} \cup \mathcal{D}_{\text{synth}}$

16:     **return** $\mathcal{D}_{\text{aug}}$

17: **end procedure**

---

out either from a pre-trained segmentation model or, when available (like in the CARLA simulator), directly from provided semantic maps. Once we have the semantic layout, image synthesis proceeds as in the segmentation case.

## 4. Results

Our experiments aim to analyze whether training models on augmented data boosts overall segmentation and autonomous driving (AD) performance. The results for subgroup-specific performance improvements are mentioned in the Supplementary Material. We use two segmentation datasets: Waymo Open Dataset [34] (37,618 annotated images) and BDD100K [41] (7,000 annotated images). Our operational design domain includes weather (Rainy, Clear, Cloudy) and time of day (Dawn/Dusk, Morning, Night), yielding 9 subgroups (see Appendix). For AD, we use a CARLA simulator-based dataset with an expert driving policy [8]. Data is collected across 3 towns, 15 routes, with fixed weather per route for realism. Testing involves 27 routes under all weather conditions, using the same operational design domain as segmentation.

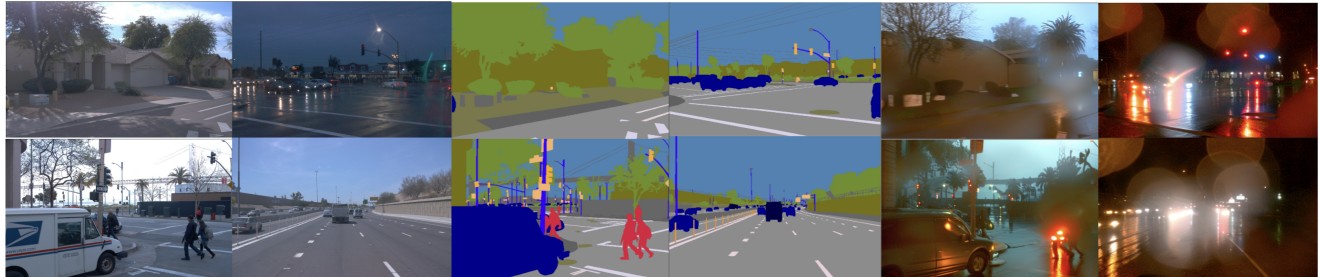

Ground Truth      Segmentation Mask     Synthetic Image for Rainy and Night

Figure 3. **Qualitative visualization of synthetic data for an under-represented subgroup – Rainy and Night**
.

| Dataset | Model (Backbone) | Data | mIOU |
|---|---|---|---|
| Waymo | Mask2Former (R50) | Real | 46.7 |
| | Mask2Former (R50) | Real + Synth | 47.8 |
| | Mask2Former(Swin-T) | Real | 50.2 |
| | Mask2Former (Swin-T) | Real + Synth | **51.5** |
| | SegFormer (MIT-B3) | Real | 45.4 |
| | SegFormer (MIT-B3) | Real + Synth | 46.2 |
| BDD100K | Mask2Former (R50) | Real | 51.5 |
| | Mask2Former (R50) | Real + Synth | 55.1 |
| | Mask2Former (Swin-T) | Real | 57.2 |
| | Mask2Former (Swin-T) | Real + Synth | **58.6** |
| | SegFormer (MIT-B3) | Real | 53.9 |
| | SegFormer (MIT-B3) | Real + Synth | 54.9 |

Table 1. **Models trained on augmented data outperform those trained on original data for segmentation tasks**.

## 4.1. Implementation Details

We fine-tune ControlNet on the Waymo, BDD (segmentation), and CARLA (autonomous driving) datasets. Both images and masks are resized to $512 \times 512$. We use a 5-epoch training regime, a learning rate of $10^{-5}$, and 8 A-5000 GPUs. During inference, Algorithm 1 generates synthetic data for under-represented subgroups through style swapping as discussed earlier. For semantic segmentation experiments we use Mask2Former (Swin-T [23] and ResNet-50 backbones [14]) and SegFormer (MIT-B3 [38] backbone). For E2E autonomous driving experiments, we fine-tune NEAT, AIM-2D, and AIM-BEV [8] on a single A-5000 GPU for model training/evaluation.

## 4.2. Effect of Fine-tuning on final performance

We analyze segmentation models using standard metrics (mIoU, mean Accuracy, mean DICE, mean F1), calculated across classes in the Waymo and BDD100K datasets. For AD, we employ Route Completion, Infraction Score, and Driving Score [8, 9]. Synthetic data augmentation improves segmentation performance as shown in Table 1. Notable gains include +1.3 mIoU for Mask2Former (Swin-T) on

| Model | Data | Driving Score (↑) |
|---|---|---|
| NEAT [8] | Real | 20.3 |
| NEAT [8] | Real + Synth | 13.63 |
| AIM-2D [9] | Real | 21.64 |
| AIM-2D [9] | Real + Synth | 34.36 |
| AIM-BEV [9] | Real | 28.61 |
| AIM-BEV [9] | Real + Synth | **35.86** |
| Expert | - | 54.6 |

Table 2. **Autonomous Driving performance improves with models trained on augmented training datasets.**

Waymo and +3.6 mIoU for Mask2Former (ResNet50) on BDD100K. In AD (CARLA simulator), AIM-2D and AIM-BEV benefit from synthetic data, primarily due to improved Infraction Scores (fewer collisions). In contrast, NEAT's performance degrades, likely due to inconsistencies in spatial geometry within synthetic multi-camera views. We believe that the above improvements are likely due to synthetic images providing context-sensitive augmentation, aiding generalization across subgroups.

## 5. Conclusion and Limitation

This paper addresses the challenge of biased datasets that hinder autonomous driving performance in under-represented conditions. Our method SynDiff-AD, provides a solution by leveraging advanced text-to-image generation with ControlNet to realistically augment datasets while preserving semantic details, eliminating manual labeling. We show the capability of SynDiff-AD to improve segmentation and autonomous driving tasks. However, a key limitation of our approach is the need for subgroups specified in natural language. Future research could discover such test subgroups for image-based datasets through image captions. Additionally, future directions include combining our generative approach with other data augmentation or task adversarial augmentation techniques and refining ControlNet to maintain spatial geometry within multi-camera views for AD. These would improve the robustness of AD systems.

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
