# Improving End-To-End Autonomous Driving with Synthetic Data from Latent Diffusion Models

## Supplementary Material

## 1. Organization

We outline the organization of the supplementary section here as follows:

1. We outline the dataset distribution across all subgroups for the datasets used in this paper in Section 2.
2. Section 3 discusses the impact of Caption Generation on the quality of synthetic data.
3. Section 4 discusses the subgroup or condition-specific performance of both semantic segmentation models and autonomous driving models fine-tuned on original and augmented datasets.
4. The performance of Autonomous Driving models (AD) over different subgroups is elaborated in Section 5.
5. Finally, we provide qualitative visualization for both segmentation and driving tasks in Section 6.

## 2. Dataset Analysis

For semantic segmentation tasks, the operation design domain $\mathcal{Z}$ and its corresponding semantic dimensions $\mathcal{Z}_{[0,1]}$ are based on weather $\in \mathcal{Z}_0 = [\text{Rainy, Clear, Cloudy}]$ and time of day $\in \mathcal{Z}_1 = [\text{Dawn/Dusk, Day, Night}]$. We present the data distribution for the reader as a reference for both the BDD datasets and the Waymo Datasets.

For autonomous driving tasks, the operation design domain $\mathcal{Z}$ and its corresponding semantic dimensions $\mathcal{Z}_{[0,1]}$ are based on weather $\in \mathcal{Z}_0 = [\text{Rainy, Clear, Cloudy}]$ and time of day $\in \mathcal{Z}_1 = [\text{Twilight, Morning, Night}]$. We present the data distribution for the reader as a reference for the expert driving data compiled through CARLA.

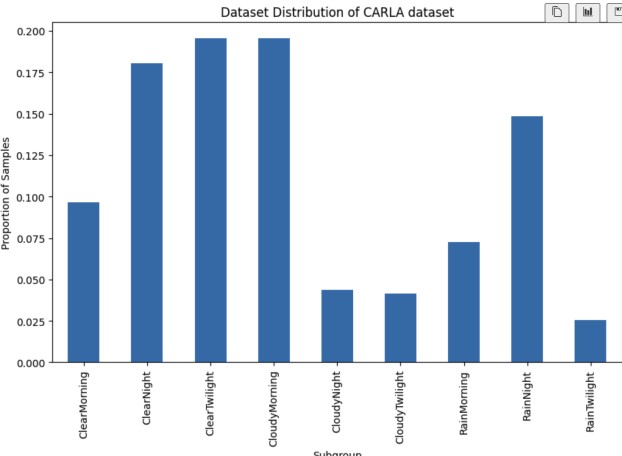

Figure 3. The distribution of autonomous driving **AD** for all identified subgroups in the training dataset.

| Distribution | BDD100K | | Waymo | |
|---|---|---|---|---|
| | CaG | no CaG | CaG | no CaG |
| Clear, Day | **162.16** | 202.02 | 152.74 | **146.99** |
| Clear, Dawn/Dusk | **66.47** | 67.05 | **150.92** | 160.57 |
| Clear, Night | **211.45** | 273.28 | **46.77** | 80.84 |
| Cloudy, Day | **134.24** | 148.49 | 118.51 | **114.93** |
| Cloudy, Dawn/Dusk | **144.94** | 199.48 | **214.55** | 224.94 |
| Cloudy, Night | **152.65** | 246.72 | **58.12** | 107.57 |
| Rainy, Day | **133.96** | 154.72 | 121.69 | **102.79** |
| Rainy, Dawn/Dusk | **199.83** | 229.45 | **124.35** | 129.68 |
| Rainy, Night | **291.66** | 349.22 | **62.21** | 112.75 |

Table 1. **Comparison of FD with and without caption generation for both datasets.** We show comprehensively that the caption generation reduces the FD score on CLIP-VIT-L16 features between the generated and the ground truth images.

## 3. Impact of Caption Generation(CaG)

To assess the impact of the proposed caption generation scheme we evaluate the quality of the synthetic images against the original ground truth images. As such we use Frechet Distance (FD) [? ] scores as a suitable benchmark for the evaluation. FD score computes the distance between the feature distributions of synthetic and original images. We compute the FD scores between the data subgroup-specific distributions for both synthetic and ground truth images. Our computation of the FD is done over the image features extracted by CLIP-VIT-L16 which has a feature dimension of size 768. Given that our caption generation scheme using VLM improves zero shot synthetic data generation with lower FD scores, we, therefore justify the use of LLaVA to caption images for text descriptions that are used in the downstream fine-tuning of ControlNet with frozen Stable Diffusion weights for semantic segmentation and AD tasks.

## 4. Effect of Fine-tuning on condition-specific performance

This set of experiments compares the effect of fine-tuning over synthetic data generated for various under-represented data subgroups. We refer you to Table 2 and Table 3 for these results.

For semantic segmentation specifically, we conduct an ablation study over two components of our proposed pipeline. First, we conduct an ablation over the effect

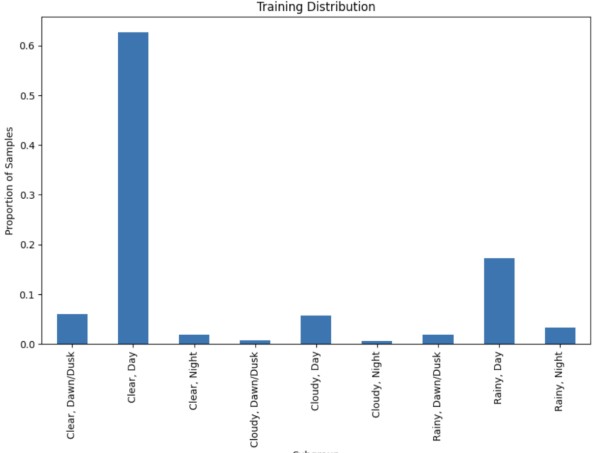
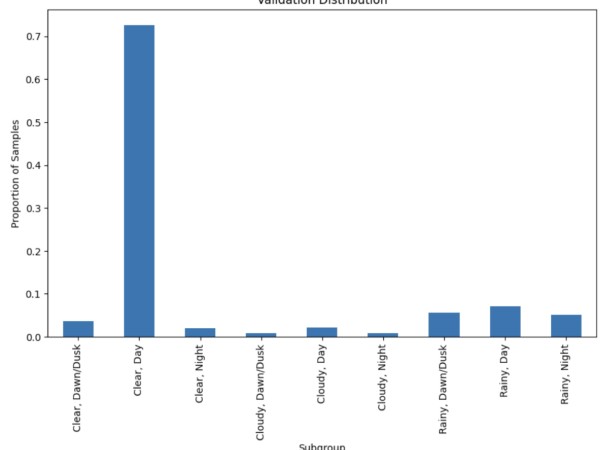

Figure 1. The distribution of image-segmentation mask pairs over all identified subgroups in the **Waymo** training and validation dataset.

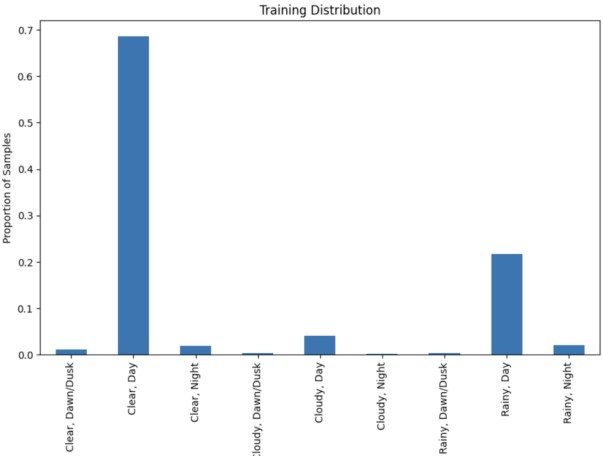
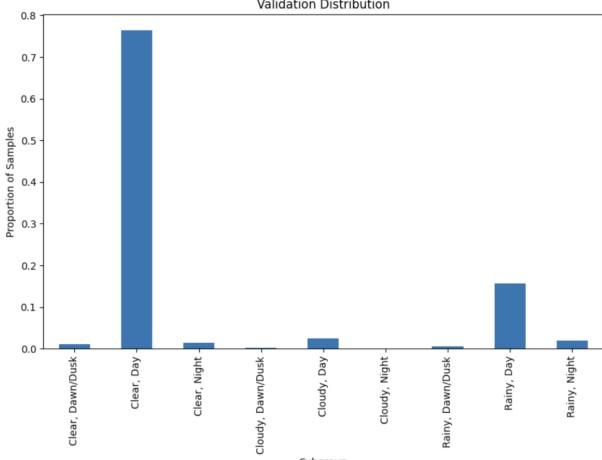

Figure 2. The distribution of image-segmentation mask pairs for all identified subgroups in the **BDD** training and validation dataset.

of fine-tuning ControlNet with data sampled over all subgroups equivalently. Thus, image-caption-segmentation mask tuples that are from rarer subgroups are sampled more selectively during fine-tuning. Synthetic data generated from this variant is referred to as Synthetic-RST (Rare Sub-Group Training). Second, we modify the method by which we sample source images for which we want synthetic data variants. Here synthetic images are sampled such that all semantic categories are equivalently present. This results in synthetic data with equivalent semantic class distributions that would enable selective training over rare semantic classes. This was shown to improve the performance of semantic segmentation models in prior work [**?**]. We report the results of the ablation study for the best-performing model i.e. Mask2Former over all the synthetic datasets. We see that across different subgroups, the best-performing models are obtained by fine-tuning over datasets augmented with synthetic data.

We report the per subgroup performance of various AD models for our tests on Autonomous Driving. In the case of Autonomous Driving, synthetic data is generated for all camera views across an entire route. Hence, the ablations proposed for semantic segmentation don't extend to AD in our setup. The averaged driving scores are reported for all 9 data subgroups for all models fine-tuned on both original dataset and augmented datasets. We see noticeable improvements in the driving score of AD models AIM-2D and AIM-BEV when trained with synthetic data augmentations using SynDiff-AD on all data subgroups. In contrast, synthetic data augmentations degraded NEAT's performance due to reasons mentioned in Section **??**.

## 5. Performance of Autonomous Driving Models

We present a detailed breakdown of the Driving Scores(**DS**) as referenced in the main paper. Here we present the Route

Completion (**RC**) scores and the Infraction Scores (**IS**) of the learned AD policies for each data subgroup and model. In the following tables, we report the above metrics for each AD model trained on the original and synthetic data. We highlight the best performing models for each metric across each subgroup. We refer you to Tables 4, 5 and 6

## 6. Qualitative Visualizations

We attempt to provide qualitative visualizations of the obtained synthetic images for different tasks and datasets. Here we sample an image and semantic mask pair and showcase its variants across different data subgroups.

### 6.1. Waymo

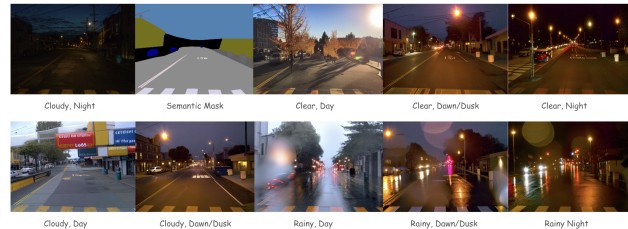

Figure 4. Sample visualization of synthetic images for a source image and mask taken in cloudy weather and night time

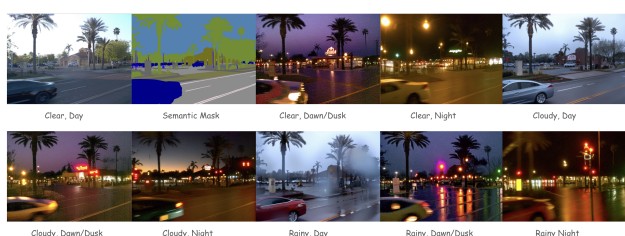

Figure 5. Sample visualization of synthetic images for a source image and mask taken in clear weather and day time

### 6.2. BDD100K

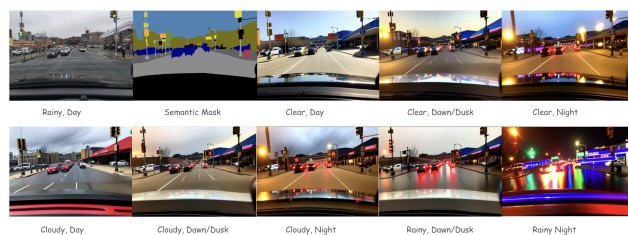

Figure 6. Sample visualization of synthetic images for a source image and mask taken in rainy weather and day time

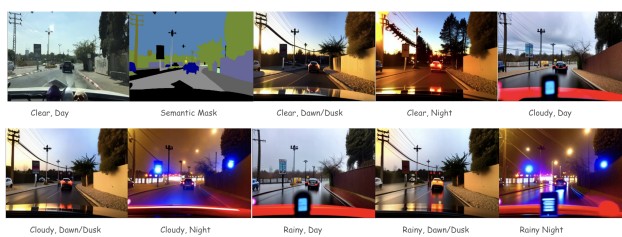

Figure 7. Sample visualization of synthetic images for a source image and mask taken in clear weather and day time

### 6.3. Autonomous Driving Carla

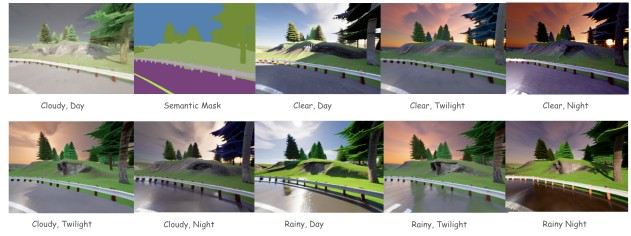

Figure 8. Sample visualization of synthetic images for a source image and mask taken in cloudy weather and day time

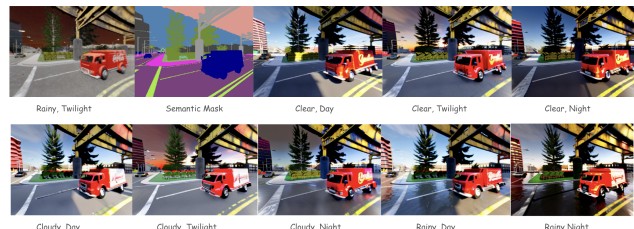

Figure 9. Sample visualization of synthetic images for a source image and mask taken in rainy weather and twilight time

| Dataset | Sub-Group | Original | Synthetic | Synthetic RST | Synthetic CEQ | Synthetic RST-CEQ |
|---|---|---|---|---|---|---|
| Waymo | Clear, Dawn/Dusk | 42.2 | 44.5 | 43.1 | 43.8 | **46.6** |
| | Clear, Day | 51.8 | **52.9** | 51.0 | 51.6 | 52.1 |
| | Clear, Night | 33.6 | 36.4 | **39.3** | 34.2 | 33.9 |
| | Cloudy, Dawn/Dusk | 48.2 | 48.8 | 47.7 | **49.5** | 49.2 |
| | Cloudy, Day | **56.3** | 55.8 | **56.3** | 52.5 | 55.4 |
| | Cloudy, Night | 38.3 | 38.1 | 35.9 | 37.5 | **39.9** |
| | Rain, Dawn/Dusk | 39.8 | **41.4** | 41.4 | 40.8 | 41.0 |
| | Rain, Day | 50.7 | 50.7 | **52.7** | 51.5 | 50.8 |
| | Rain, Night | 35.1 | 34.8 | 35.1 | **36.5** | 36.1 |
| BDD100K | Clear, Dawn/Dusk | 42.3 | 51.1 | **52.3** | 50.6 | 47.8 |
| | Clear, Day | 56.7 | 57.5 | 57.2 | 55.9 | **57.8** |
| | Clear, Night | 42.0 | **51.3** | 42.7 | 49.0 | 45.8 |
| | Cloudy, Dawn/Dusk | 40.8 | 42.4 | 42.6 | 35.8 | **44.3** |
| | Cloudy, Day | 52.2 | **60.4** | 55.8 | 57.0 | 59.6 |
| | Cloudy, Night | 35.4 | 49.3 | 47.8 | 49.0 | **52.3** |
| | Rain, Dawn/Dusk | 49.9 | **57.6** | 52.0 | 50.1 | 49.3 |
| | Rain, Day | 54.8 | 56.6 | 56.0 | **57.7** | 55.2 |
| | Rain, Night | 30.5 | 35.8 | 35.7 | 35.2 | **36.4** |

Table 2. **Improved performance over different data subgroups with synthetic data augmentation**. We conduct an ablation study that constructs synthetic datasets using three approaches - RST, CEQ, and RST - CEQ. RST datasets comprise images from a fine-tuned ControlNet that equally samples rare subgroups during training. CEQ datasets are sampled so that the synthetic dataset's semantic class distribution is uniform. RST-CEQ incorporates both these strategies.

| Model | Aug | Clear | | | Cloudy | | | Rain | | |
|---|---|---|---|---|---|---|---|---|---|---|
| | | Twi | Day | Night | Twi | Day | Night | Twi | Day | Night |
| **NEAT** | No | 35.86 | 5.09 | 21.87 | 17.71 | 36.66 | 17.46 | 3.49 | 23.10 | 21.72 |
| **NEAT** | Yes | 12.14 | 16.73 | 8.86 | 15.95 | 14.18 | 20.24 | 6.86 | 14.48 | 13.32 |
| **AIM-2D** | No | 19.69 | 23.20 | 6.11 | **37.25** | 18.77 | 43.72 | 14.68 | 23.93 | 3.42 |
| **AIM-2D** | Yes | 39.04 | 40.30 | **29.02** | 19.11 | 33.44 | **46.32** | 16.68 | 50.94 | **34.23** |
| **AIM-BEV** | No | 39.78 | 31.42 | 2.73 | 29.88 | **44.68** | 43.22 | 18.07 | 43.73 | 3.97 |
| **AIM-BEV** | Yes | **58.37** | **47.94** | 25.42 | 14.93 | **44.22** | 35.03 | **27.52** | **53.67** | 15.64 |

Table 3. **AD models trained on augmented datasets exhibit improved driving performance** We show that AD models fine-tuned on augmented datasets (indicated by Aug) have improved performance, especially over rare subgroups where the models trained on the original dataset underperform.

| Metric | Aug | Clear | | | Cloudy | | | Rain | | |
|---|---|---|---|---|---|---|---|---|---|---|
| | | Twi | Day | Night | Twi | Day | Night | Twi | Day | Night |
| **RC** | | 53.08 | 43.29 | 28.44 | 35.48 | 53.75 | 53.41 | 9.22 | 53.24 | 33.81 |
| **IS** | No | **0.828** | 0.386 | 0.747 | 0.629 | **0.829** | 0.499 | **0.595** | **0.645** | **0.667** |
| **DS** | | 35.86 | 5.09 | 21.87 | 17.71 | 36.66 | 17.46 | 3.49 | 23.10 | 21.72 |
| **RC** | | 33.56 | 33.47 | 52.21 | 32.97 | 33.22 | 50.73 | 31.10 | 33.63 | 30.03 |
| **IS** | Yes | 0.597 | 0.644 | 0.441 | **0.654** | 0.631 | 0.667 | 0.453 | 0.633 | 0.649 |
| **DS** | | 12.14 | 16.73 | 8.86 | 15.95 | 14.18 | 20.24 | 6.86 | 14.48 | 13.32 |

Table 4. **Performance of NEAT across different data sub-groups**

| Metric | Aug | Clear | | | Cloudy | | | Rain | | |
|---|---|---|---|---|---|---|---|---|---|---|
| | | Twi | Day | Night | Twi | Day | Night | Twi | Day | Night |
| RC | | 76.04 | 55.22 | 84.91 | 54.05 | 76.05 | 55.02 | 48.65 | 84.61 | 45.97 |
| IS | No | 0.224 | 0.483 | 0.073 | 0.729 | 0.204 | 0.727 | 0.259 | 0.244 | 0.352 |
| DS | | 19.69 | 23.20 | 6.11 | **37.25** | 18.77 | 43.72 | 14.68 | 23.93 | 3.42 |
| RC | | 84.81 | **55.30** | 84.05 | 54.98 | 83.59 | 55.02 | 82.58 | 77.02 | **69.38** |
| IS | Yes | 0.392 | 0.631 | 0.312 | 0.339 | 0.373 | **0.791** | 0.266 | 0.551 | 0.472 |
| DS | | 39.04 | 40.30 | **29.02** | 19.11 | 33.44 | **46.32** | 16.68 | 50.94 | **34.23** |

Table 5. **Performance of AIM-2D across different data-subgroups**

| Metric | Aug | Clear | | | Cloudy | | | Rain | | |
|---|---|---|---|---|---|---|---|---|---|---|
| | | Twi | Day | Night | Twi | Day | Night | Twi | Day | Night |
| RC | | 83.48 | 55.18 | 76.31 | **55.18** | **100.0** | **55.18** | 64.51 | 83.14 | 69.21 |
| IS | No | 0.436 | 0.589 | 0.038 | 0.573 | 0.447 | 0.706 | 0.269 | 0.462 | 0.255 |
| DS | | 39.78 | 31.42 | 2.73 | 29.88 | **44.68** | 43.22 | 18.07 | 43.73 | 3.97 |
| RC | | **100.0** | **55.28** | **100.0** | 23.66 | 90.86 | 36.99 | **85.92** | **100.0** | 69.38 |
| IS | Yes | 0.584 | **0.706** | 0.254 | 0.438 | 0.452 | 0.677 | 0.374 | 0.536 | 0.285 |
| DS | | **58.37** | **47.94** | 25.42 | 14.93 | **44.22** | 35.03 | **27.52** | **53.67** | 15.64 |

Table 6. **Performance of AIM-BEV across different data-subgroups**