# OpenReview forum: "Improving End-To-End Autonomous Driving with Synthetic Data from Latent Diffusion Models"
_thecvf.com/CVPR/2024/Workshop/VLADR — VLADR 2024 Poster_

### Official Review · Reviewer_BNM4 · 2024-04-17

**Rating:** 6
**Confidence:** 4

**Review:**

Overall Review:
This paper proposes a new method for generating extra training data from existing datasets in autonomous driving scenarios. By leveraging existing LDM and VLM, the method can produce diverse and challenging samples with no human effort. The effectiveness of this method is demonstrated through experiments on semantic segmentation and end-to-end autonomous driving tasks.

Strengths:
- The paper studies an intriguing issue: using generative models to create more diverse training data, which aids in training perception models.
- The paper is easy to follow and the experiments are well conducted.
- The quantitative results are comprehensive and qualitative results are convincing.

Weaknesses:
- The proposed idea is interesting. However, conditioning on semantic segmentation sometimes fails to guide the LDM to produce realistic and physically consistent samples, as shown by some qualitative results.
- The selected tasks are limited. It would be beneficial to validate on more challenging tasks, such as 3d object detection.
- The ablation study on the ratio of synthesized data to real data is missing, which would provides more insights on how to use the synthesized data effectively.

Rating: Marginally above acceptance threshold.

---

### Decision · Program_Chairs · 2024-04-22

Accept (Poster)